# Photocatalytic phosphine-mediated water activation for radical hydrogenation

Jingjing Zhang[1], Christian Mück-Lichtenfeld[1,2] & Armido Studer[1✉]

The chemical activation of water would allow this earth-abundant resource to be transferred into value-added compounds, and is a topic of keen interest in energy research[1,2]. Here, we demonstrate water activation with a photocatalytic phosphine-mediated radical process under mild conditions. This reaction generates a metal-free $PR_3$–$H_2O$ radical cation intermediate, in which both hydrogen atoms are used in the subsequent chemical transformation through sequential heterolytic ($H^+$) and homolytic ($H^·$) cleavage of the two O–H bonds. The $PR_3$–OH radical intermediate provides an ideal platform that mimics the reactivity of a 'free' hydrogen atom, and which can be directly transferred to closed-shell π systems, such as activated alkenes, unactivated alkenes, naphthalenes and quinoline derivatives. The resulting H adduct C radicals are eventually reduced by a thiol co-catalyst, leading to overall transfer hydrogenation of the π system, with the two H atoms of water ending up in the product. The thermodynamic driving force is the strong P=O bond formed in the phosphine oxide by-product. Experimental mechanistic studies and density functional theory calculations support the hydrogen atom transfer of the $PR_3$–OH intermediate as a key step in the radical hydrogenation process.

Water can be transformed to valuable hydrogen gas ($H_2$) through serial chemical processes, which allow it to be applied as a potential transfer hydrogenation reagent in the reduction of unsaturated compounds[1,2]. However, the main challenges for water activation originate from two features (Fig. 1a). (1) Whereas deprotonation of water ($pK_a = 15.74$) can be readily achieved with an appropriate base, homolytic cleavage is challenging due to the high thermodynamic stability of $H_2O$ as a result of the high bond dissociation energy (BDE = 118 kcal mol⁻¹) of the O–H bonds[3,4]. (2) Further, the OH radical (hydroxyl radical), if generated, cannot provide the second hydrogen atom, as this would lead to a high-energy oxygen atom[5]. As a result, catalysts or mediators are required, and current approaches to water activation follow three different strategies: oxidative addition activation[6–8], metal–ligand cooperation activation[9] and coordination-induced bond weakening[10–12] (Fig. 1b). The first two approaches mainly focus on the use of *d*-block transition metals, and transformations dealing with main group elements remain very rare. Several examples capitalizing on the concept of coordination-induced bond weakening through generation of the corresponding $H_2O$ adducts have been disclosed using transition metal complexes or main group elements with empty *d* or *p* orbitals[10]. However, as $H_2O$ adduct formation also increases the acidity, the majority of such $H_2O$–complexes react as proton donors. Considering the homolytic O–H bond cleavage, a few examples are known, with samarium(II)[13–18], titanium(III)[19–21], borane[22–24] and bismuth(II)[25] compounds, among others, enabling homolytic O–H bond cleavage processes. These activated water complexes have been shown to reduce reactive radicals, leading to closed-shell products, in which hydrogen atom transfer (HAT) is thermodynamically favoured (Fig. 1c)[10,19–26]. By contrast, examples of intermolecular HAT

from $H_2O$–intermediates to closed-shell π systems leading to reactive H adduct C radicals, especially considering metal-free $H_2O$ adduct intermediates, remain desirable. Notably, in situ-generated Fe, Mn and Co hydride complexes have been intensively investigated as HAT reagents in the reaction with alkenes to give the corresponding H adduct C radicals, in so-called metal hydride hydrogen atom transfers (MHAT)[27–29]. In these MHAT processes, the H atom to be delivered is generally derived from a hydride source, such as a silane. Intermolecular HAT to π systems from activated water are known for $SmX_2$–$H_2O$ adducts, albeit with limited scope, and metal-free variants need to be uncovered[30–33].

Considering transition metal-free variants, there are several challenges to be overcome. The currently known water adducts that are able to act as HAT reagents are relatively stable, so that hydrogen transfer to π systems leading to reactive C radicals is endergonic. As a consequence, except in the examples with Sm(II)[30–36], only HAT to reactive radicals is established with such species. Furthermore, the $H_2O$-based H donors should be readily generated, be sufficiently reactive to engage in thermodynamically favoured HAT to π systems, but at the same time must have a sufficient lifetime, which renders reagent design highly challenging. It is remarkable that a few examples of metal-free radical H atom transfers to realize hydrogenation of closed-shell π systems have been reported. Thus, 9,10-dihydroanthracene, xanthene or tetralin were shown by Rüchardt and co-workers to transfer an H atom to closed-shell π systems[37–39]. However, as these H donors themselves are stable closed-shell compounds, intermolecular HAT in a retrodisproportionation reaction is strongly endothermic and kinetically not favoured. Therefore, these processes are very narrow in scope and only occur at temperatures above 300 °C.

[1]Organisch-Chemisches Institut, Westfälische Wilhelms-Universität, Münster, Germany. [2]Center for Multiscale Theory and Computation, Westfälische Wilhelms-Universität, Münster, Germany. ✉e-mail: studer@uni-muenster.de

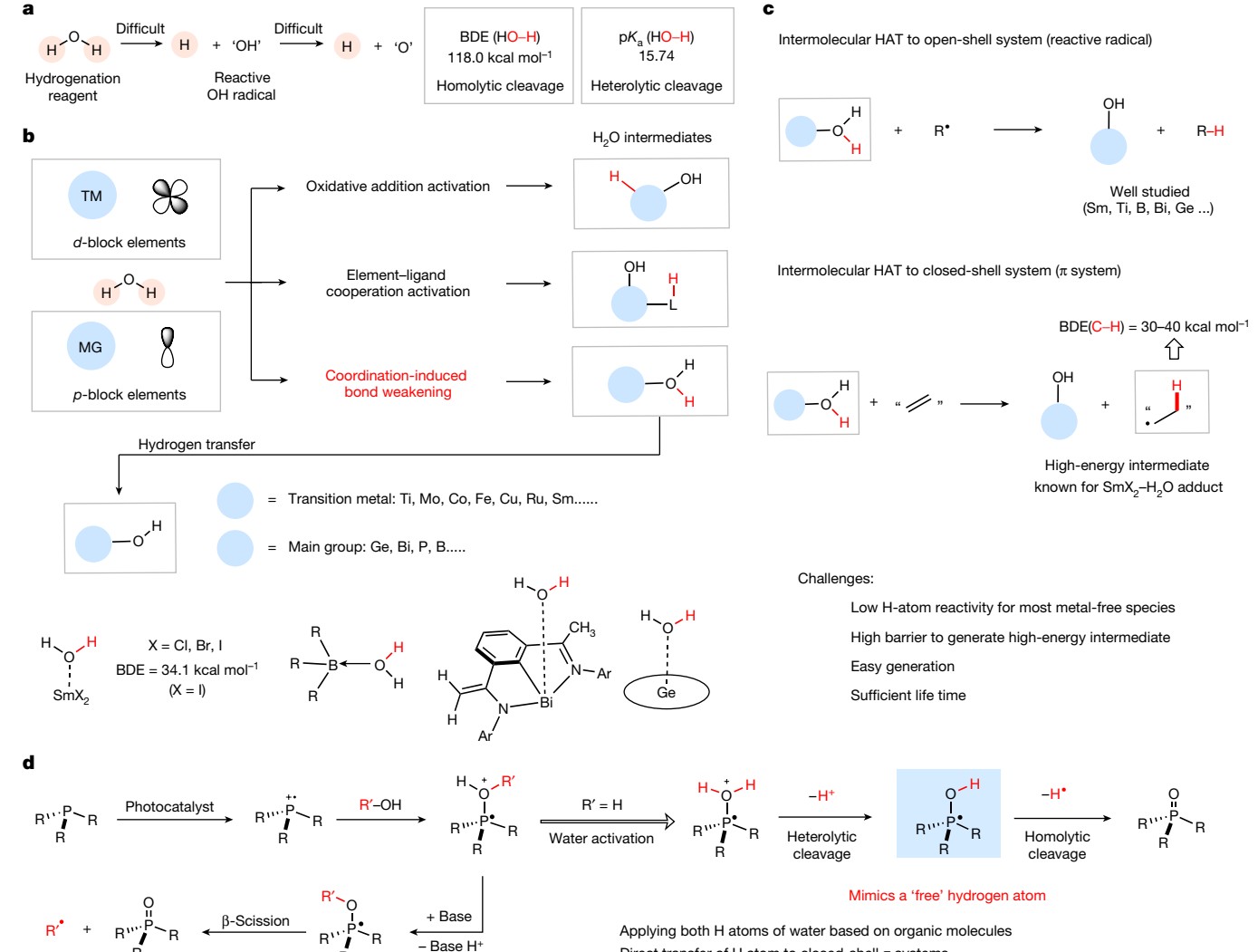

**Fig. 1 | Background and concept. a**, Water as a potential transfer hydrogenation reagent; BDE and p$K_a$. **b**, Overview of different water activation modes and examples for activated $H_2O$ adducts. **c**, Radical reductions through intermolecular HAT from water adducts and HAT to a closed-shell π system.

Challenges associated with the latter reactivity mode. **d**, This work: photocatalytic phosphine-mediated water activation to provide a 'free' H atom through sequential heterolytic and homolytic cleavage of two O–H bonds in the activated water adduct. MG, main group; TM, transition metal.

Considering that phosphine radical cations can act as tunable mediators in photoredox catalysis for the generation of carbon- and heteroatom-centred radicals through deoxygenation (Fig. 1d)[40,41], we designed a related approach for water activation by interaction of a phosphine radical cation through its singly occupied molecular orbital with water, using photoredox catalysis. The $PR_3$–$H_2O$ radical cation can be deprotonated, generating a $PR_3$–OH radical intermediate in a heterolytic O–H bond cleavage, as previously suggested by Pandey and co-workers[42]. The strong BDE of the P=O bond in phosphine oxides ensures a high thermodynamic driving force for the homolytic cleavage of the second O–H bond[43], enabling the $PR_3$–OH intermediate to behave like a 'free' H atom, providing the opportunity to explore H atom radical chemistry.

To experimentally validate the reaction design and H atom reactivity of the proposed $PR_3$–OH intermediate, triphenylphosphine (PPh$_3$)-mediated hydrogen evolution was investigated first. Pleasingly, with [Ir(dF(CF$_3$)ppy)$_2$(dtbbpy)](PF$_6$)] (dF(CF$_3$)ppy, 2-(2,4-difluorophenyl)-5-trifluoromethylpyridine; dtbbpy, 4,4′-di-*tert*-butyl-2,2′-bipyridine, **PC1**) as the photocatalyst and $H_2O$ as the hydrogen source in acetonitrile under irradiation with a 5 W blue light-emitting diode (LED), generation of $H_2$ was experimentally verified in both solution and gas phase, with concomitant formation of

triphenylphosphine oxide as the by-product (Fig. 2a). This important experiment indicated that the proposed radical PPh$_3$–OH intermediate can be generated, and further that it has a sufficient lifetime so that intermolecular reactions with such an H donor become feasible, as shown by the successful generation of $H_2$. Encouraged by this finding, transfer hydrogenation of unactivated alkenes with water as the H donor was attempted (Fig. 2b). According to the proposal, the first H atom of the activated $H_2O$ is released as a proton (Fig. 1d), and therefore we decided to use a thiol co-catalyst that is able to transform the proton donor (PPh$_3$–OH$_2^+$) into a radical H donor through thiolate protonation. The thiol in turn is capable of efficiently reducing an alkyl radical. The key species for the designed process is the suggested $PR_3$–OH intermediate, which should show sufficient reactivity to transfer its H atom intermolecularly to an unactivated alkene.

4-Allyltoluene (**1a**) was selected as the model substrate to evaluate the feasibility of the targeted radical alkene transfer hydrogenation. Pleasingly, using tris(4-methoxyphenyl)phosphine **P1** as mediator, **PC1** as photocatalyst, 2,4,6-triisopropylbenzenethiol (**HAT1**) as HAT catalyst and $H_2O$ as hydrogen source in acetonitrile under blue LED irradiation, the hydrogenation product **2a** could be obtained in 86% isolated yield at 20 °C (Fig. 2c, entry 1). Other triaryl phosphines **P2**–**P6** provided

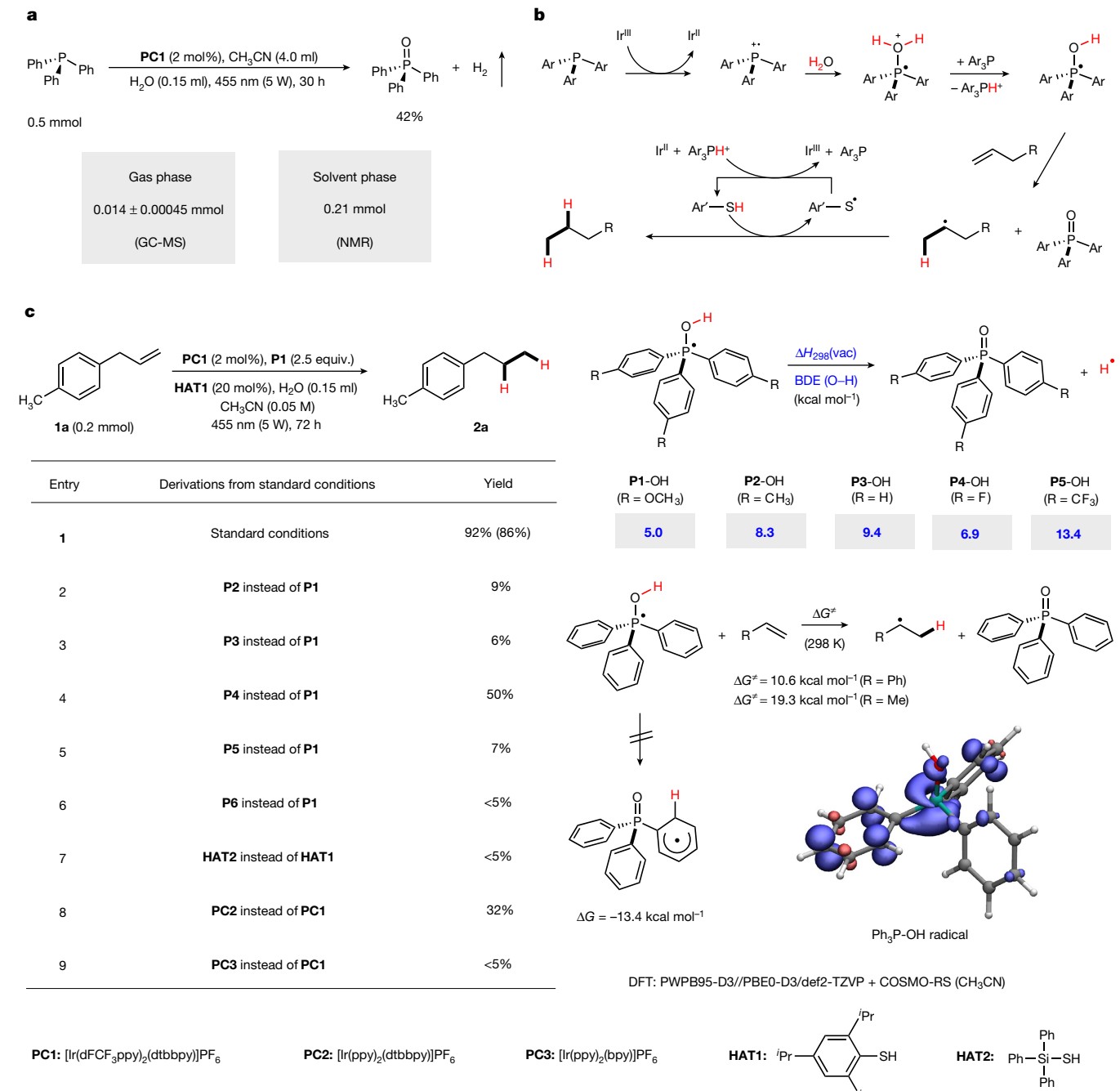

**Fig. 2 | Reaction design and mechanism analysis. a**, Hydrogen evolution through photocatalytic triphenylphosphine-mediated water activation. **b**, General mechanism for HAT of PR₃–OH intermediates to closed-shell unactivated alkenes. **c**, Reaction optimization for transfer hydrogenation of **1a** and BDE of the O–H bond in various PR₃–OH intermediates, obtained by density functional theory (DFT) calculations. Yields were determined by gas chromatography–mass spectrometry (GC–MS) with tetradecane as the internal standard. Calculated activation energy for the intermolecular HAT from the **P3**–OH intermediate to styrene and propene as model reaction. The structure displays the spin density distribution of the Ph₃P–OH radical indicating delocalization of the radical spin from the phosphorus atom into the adjacent phenyl ring. **P6**, bis(4-methoxyphenyl)(methyl)phosphine. $\Delta G$, reaction free energy (298 K); $\Delta G^{\neq}$, free energy barrier (298 K); $\Delta H_{298}$(vac), enthalpy of bond dissociation.

worse results (Fig. 2c, entries 2–6), and hydrogenation failed with the thiol co-catalyst **HAT2** (Fig. 2c, entry 7). Other frequently used Ir-based photocatalysts (**PC2**, **PC3**) provided lower yields (Fig. 2c, entries 8 and 9) and all reagents are required for successful hydrogenation of **1a** (Supplementary Information).

To get an idea regarding the structure and the H atom affinity of the H donor, density functional theory (DFT) calculations were performed (Fig. 2c and Supplementary Information). First, the O–H bond dissociation energy of different PAryl₃–OH intermediates was calculated

to be very low for all compounds considered, ranging from 5.0 to 13.4 kcal mol⁻¹. We note that the lowest bond energies are obtained for the H atom donors formed with **P1** and **P4**, which show the highest yields in the hydrogenation of **1a**. Interestingly, we found that transfer of the H atom to the *ortho*-position of the aryl substituent leads to a more stable structure (shown for **P3**). However, due to the non-linearity of that intramolecular HAT, we did not find any transition state for such a process and therefore this cyclohexadienyl radical was not further considered as the H atom donor in these reactions.

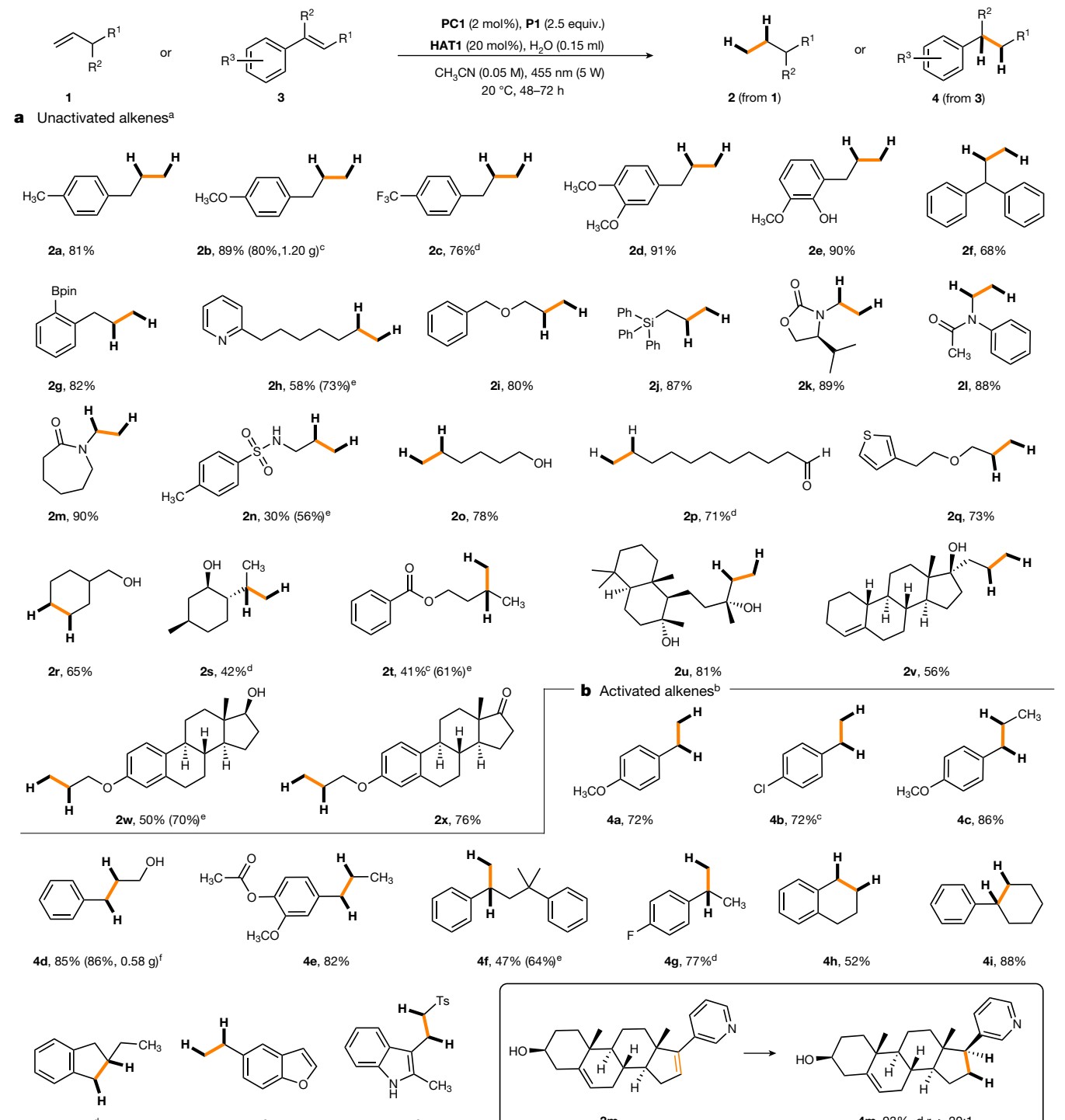

**Fig. 3 | Transfer hydrogenation substrate scope. a**, The reduction of unactivated alkenes. **b**, The reduction of styrene derivatives. Yields given correspond to isolated yields. See Supplementary Information for specific reduction conditions that deviate from the standard conditions. Diastereoselectivity was determined by [1]H NMR spectroscopy. [a]Triaryl phosphine **P1** as the mediator in combination with 0.2 equivalent of **HAT1**.

[b]Triaryl phosphine **P3** as the mediator in combination with 0.15 equivalent of **HAT1**. [c]Yield obtained for gram-scale experiment (10.0 mmol scale). [d]GC yield with tetradecane as internal standard. [e]Reaction conducted at 40 °C. [f]Yield for gram-scale experiment (5.0 mmol scale). Bpin, boronic acid pinacol ester; d.r., diastereomeric ratio; Ts, tosyl.

Intermolecular HAT of the **P3**-derived PAryl₃–OH radical intermediate was calculated for styrene and propene as the H atom acceptors. In agreement with the experimental findings, HAT to propene is feasible at room temperature with a free energy barrier of 19.3 kcal mol⁻¹. As expected, due to the radical stabilization of the styryl radical, an even lower barrier was found for HAT to styrene (10.6 kcal mol⁻¹).

With the optimized conditions in hand, various unactivated alkenes bearing different functionalities could be efficiently hydrogenated. Allylarenes carrying electron-donating or -withdrawing substituents at the arene moiety provided the corresponding propylarenes **2a**–**2f** with high yields (Fig. 3a). The radical hydrogenation was compatible with commonly used functional groups, including boronic acid pinacol

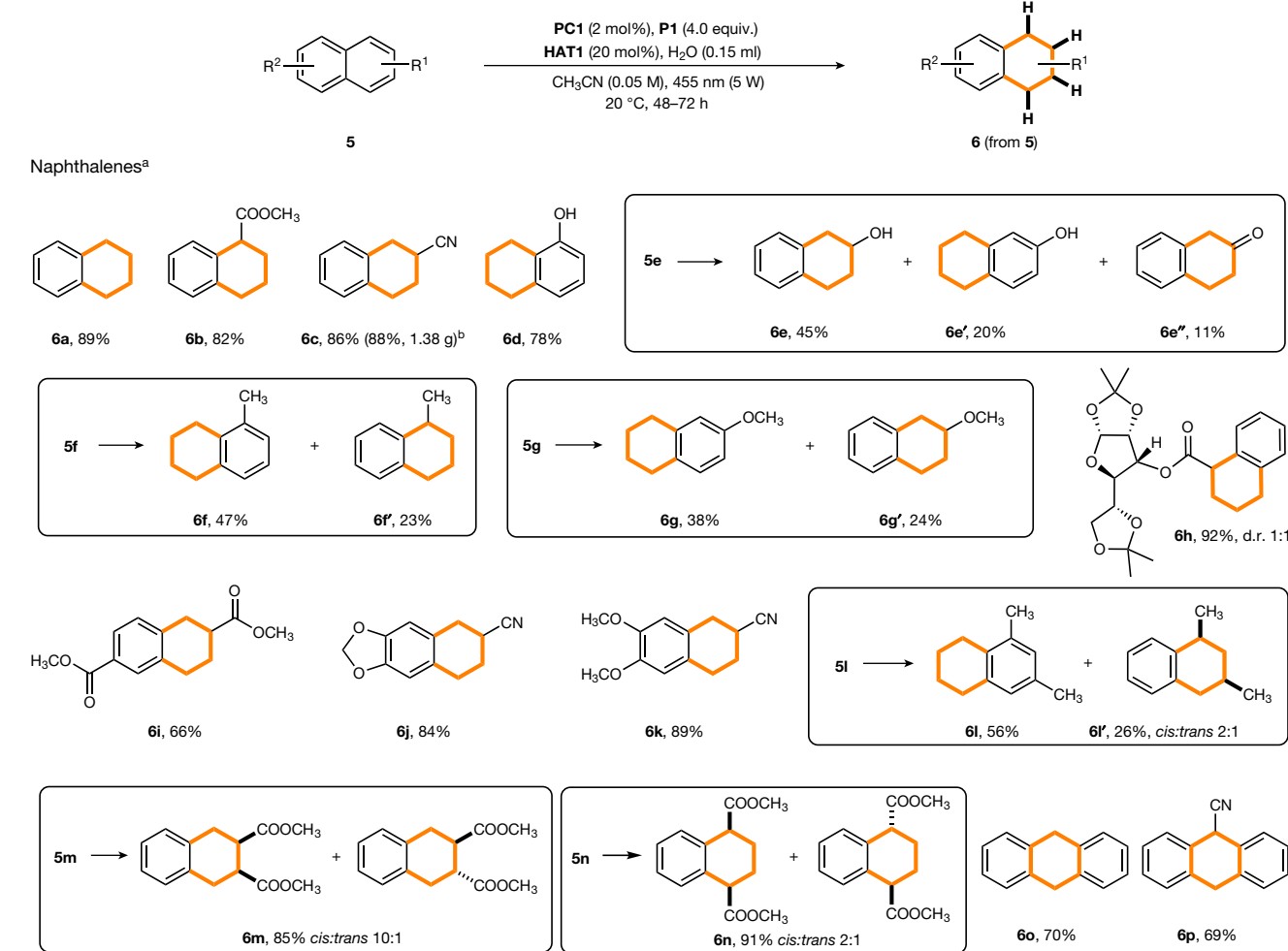

**Fig. 4 | The reduction of naphthalenes.** Yields given correspond to isolated yields. See Supplementary Information for specific reduction conditions that deviate from the standard conditions. Diastereoselectivity was determined by [1]H NMR spectroscopy. [a]Triaryl phosphine **P1** as the mediator in combination with 0.2 equivalent of **HAT1**. [b]Yield obtained for gram-scale experiment (10.0 mmol scale).

ester (**2g**), pyridine (**2h**), ether (**2i**), triphenylsilyl (**2j**), oxazolidinone (**2k**), amide (**2l** and **2m**), sulfonamide (**2n**), free alcohols (**2o** and **2r**), aldehyde (**2p**) and thiophene (**2q**) moieties. Disubstituted alkenes could also be reduced, albeit with moderate yields (**2s** and **2t**). Our mild protocol also permits late-stage hydrogenation of more complex compounds bearing a terminal unactivated double bond, as documented by the reduction of sclareol (**2u**), allylestrenol (**2v**), β-estradiol (**2w**) and an estrone derivative (**2x**). Notably, the reduction of allylestrenol **1v** furnished the product **2v** with complete chemoselectivity and the trisubstituted alkene moiety in the ring remained unreacted. The more reactive styrene derivatives could be readily hydrogenated using PPh₃ (**P3**) as the H atom mediator and reaction time could be shortened (48 h). Various styrene derivatives, including α- and β-substituted congeners, as well as trisubstituted systems, could be reduced in moderate to excellent yields (Fig. 3b, **4a**–**4l**). The steroidal compound **3m** was hydrogenated with complete chemo- and diastereoselectivity.

Hydrogenation of arenes provides an atom-economic, straightforward and efficient route for the construction of three-dimensional structures through easily accessed planar aromatic compounds. However, arene reduction through H atom transfer is more difficult than alkene hydrogenation, as the initial HAT leads to de-aromatization. As is known, radical MHAT by Mn/Co/Fe hydrides to arenes is very challenging[44,45]. Motivated by our calculations, which showed that intramolecular H transfer to the arene moiety in the PAryl₃–OH

radical intermediate is thermodynamically feasible (Fig. 2c), we next attempted the challenging hydrogenation of arenes (Fig. 4). We were very pleased to find that naphthalene (**5a**) and its derivatives with ester (**5b**), cyano (**5c**) and hydroxy (**5d**) groups could be hydrogenated to afford the corresponding tetrahydronaphthalenes with high yields. Hydrogenation of the second arene moiety was not observed in these cases, which can be understood by the larger resonance energy of the arene moiety in tetrahydronaphthalene compared to naphthalene. Naphthalenes with electron-withdrawing substituents were regioselectively de-aromatized at the substituted arene ring. Naphthalene derivatives bearing electron-donating methyl or methoxy groups (see compounds **5f** and **5g**) provided mixtures of the two possible tetrahydronaphthalenes. The regioselectivity issue was further investigated by DFT calculations (Supplementary Information). For 1-methylnaphthalene (**5f**) the activation barrier for the intermolecular HAT from the **P1**–OH radical to the 2-, 3-, 4-, 5-, 6-, 7- and 8-position was calculated. All HATs are exothermic (−17.9 to −23.5 kcal mol⁻¹) and occur with low barriers (6.3 to 9.7 kcal mol⁻¹). The lowest barrier was calculated for the HAT to the C4-position (6.3 kcal mol⁻¹), but H transfer to C8 and C5 leading to the other regioisomers showed similar barriers (6.5 and 6.9 kcal mol⁻¹, respectively). This explains why no selectivity was obtained for the hydrogenation of **5f**. The barrier for the HAT correlates well with the thermodynamic stability of the H adduct (exothermicity, C4-position: −23.5 kcal mol⁻¹, C8-position:

−23.5 kcal mol$^{-1}$, C5-position: −23.0 kcal mol$^{-1}$). With this knowledge in hand, we also calculated the exothermicity for the HAT to 1-(methoxycarbonyl)naphthalene (**5b**) and 2-cyanonaphthalene (**5c**), for which perfect regiocontrol was achieved in the hydrogenation. Again, seven regioisomeric H adducts were considered for both cases. For the ester **5b**, HAT to the C4-position was thermodynamically most favoured (−26.2 kcal mol$^{-1}$) leading to the experimentally observed regioisomer **6b**. The thermodynamically most stable H adduct in the reaction with **5c** resulted from C1 addition (−27.0 kcal mol$^{-1}$), which eventually leads to the observed isomer **6c**, revealing that the regioselectivity in these hydrogenations is mainly controlled by the stability of the de-aromatized H adduct radical.

Naphthols deserve special attention: 1-naphthol gave the phenol **6d** as a single reduction product with complete regiocontrol, whereas 2-naphthol provided three different products **6e**, **6e′** and **6e″**. Formation of the ketone **6e″** indicates that the enol formed by 1,4-reduction of the naphthalene core is a likely intermediate in the hydrogenation of **5e**. The more complex naphthalene **5h** was regioselectively hydrogenated to give **6h** as a 1:1 mixture of diastereoisomers. Hydrogenation also worked well for multisubstituted naphthalenes (**5i**–**5k**). Regioselective hydrogenation occurred at the more electron poor arene moiety (**6j** and **6k**). Disubstituted naphthalenes with both substituents at the same ring afforded the targeted tetrahydronaphthalenes as diastereoisomeric mixtures (**6l**–**6n**). As expected for 1,3-dimethylnaphthalene, **6l** was formed as a major regioisomer along with **6l′**, which was isolated as a 2:1 (*cis:trans*) diastereoisomeric mixture. Complete regiocontrol was noted for the hydrogenation of the naphthalenes **5m** and **5n**. Very good 1,2-stereocontrol was achieved for **6m** (*cis:trans* = 10:1), whereas the 1,4-stereoinduction was moderate (**6n**, 2:1). Anthracene **5o** and its cyanated derivative **5p** were reduced with complete regiocontrol at the most activated central ring. Comparable yields were obtained for the gram-scale preparation of **2b**, **4d** and **6c**, documenting the practicality of this hydrogenation process. The generated phosphine oxides could be almost quantitatively recovered (Supplementary Information) and are easily reduced to give the corresponding starting phosphines[46].

The unique H atom reducing ability of this photocatalytic phosphine-mediated water activation system, in which hydrogenation proceeds through radical intermediates, was further verified by realizing direct skeletal editing of quinolines (Fig. 5a)[47]. Such a hydrogenative rearrangement is unknown for any other radical reducing system and cannot be achieved through transition metal-mediated hydrogenation processes. Several indole derivatives could be directly constructed by a reductive ring contraction of the corresponding quinoline derivatives. The rearrangement of 2-substituted quinolines (**5q**–**s** and **5u**) afforded the corresponding 2,3-disubstituted indoles **6q**–**s** and **6u** in moderate yields. A 2,4-disubstituted quinoline (**5t**) also engaged in this transformation to give the indole **6t**, and a possible mechanism is presented in Fig. 5a. Intermediate **Int1** is generated through an initial H atom transfer of the **P1**−OH intermediate to the quinoline C(4)-position. Subsequent neophyl-type rearrangement through **Int2** leads to **Int3**, which is finally reduced by the thiol co-catalyst to provide the product indole. The suggested mechanism was supported by a deuteration experiment: replacing $H_2O$ with $D_2O$ provided the bis-deuterated indole **6u-D$_2$**, showing that the initial HAT by the **P1**−OH intermediate and also the final HAT from the thiol co-catalyst occur at the same carbon atom.

We next explored the relative reactivity of two different π systems, either by installing them into the same substrate or by running competition experiments of two π systems at low conversion (Supplementary Information). The alkene moiety in 2-vinylnaphthalene (**3n**) is more reactive than the naphthalene core, and 2-ethylnaphthalene was obtained with complete chemoselectivity using PPh$_3$ (**P3**). The same outcome was noted for the more reactive phosphine **P1** (Supplementary Information). Comparing an activated alkene with an unactivated terminal alkene in 4-allyloxystyrene, we found excellent chemoselectivity

for the hydrogenation of the styrenic double bond with the milder **P3** as mediator, and allyl (4-ethylphenyl) ether was obtained in 82% yield. However, both double bonds in 4-allyloxystyrene were reduced with the more reactive phosphine **P1**. Thus, the chemoselectivity of the hydrogenation of 4-allyloxystyrene can be controlled by tuning of H atom reactivity through variation of the PAryl$_3$ component. This is reminiscent of the reactivity tuning in organometallic hydrides through ligand variation. Considering styrene pairs that slightly differ in their electronic structure by variation of their *para*-substituents, low selectivity was noted using the milder PPh$_3$-derived H donor (Me versus CN = 1:1.4; Me versus MeO = 1:2). Steric effects play a role as β-methyl-(*trans*) and also α-methyl-substituted styrenes are less reactive than their corresponding unsubstituted styrenes (by around 2:1).

As an additional advantage of the radical-type hydrogenation over well-established Pd/Rh/Ir-catalysed processes, which generally operate with $H_2$ gas[48–50], hydrogenations with concomitant cyclization or ring opening were studied. At the same time, these investigations provided further experimental support for the radical nature of these transformations (Fig. 5b). Four different dienes **7**, **9**, **11** and **13** were subjected to the standard conditions and the targeted reductive cyclization products **8**, **10**, **12** and **14** were obtained. In all cases, hydrogenated non-cyclized derivatives were not observed, showing that the cyclization is faster than direct reduction under the applied conditions. For the Si compound **7**, 6-*endo-trig* cyclization occurred exclusively, whereas the other three systems cyclized through the 5-*exo* mode and the five-membered rings were formed with good to excellent diastereoselectivities. Vinylcyclopropane **15** reacted to produce the reduced ring-opening product **16**. As a side product, alkene **17** was isolated, probably resulting from double bond isomerization of **16**. Control experiments revealed that $H_2O$ is the exclusive hydrogen source in the hydrogenation and the solvent (acetonitrile) is not involved (Fig. 5c), indicating the potential of this radical hydrogenation for the preparation of deuterated compounds using cheap $D_2O$, as realized for the preparation of **2d-D$_2$** and **6j-D$_4$**.

Kinetic experiments, from running the reaction of **3a** in either $H_2O$ or in $D_2O$ at low conversion, indicated that the HAT process between the PR$_3$−OH intermediate and the alkene is the rate-determining step, with a kinetic isotope effect (KIE) value of 3.2 (Fig. 5c). This is further supported by an additional experiment in which hydrogenation was conducted using a 1:1 mixture of $H_2O$:$D_2O$ with a lower deuterium incorporation observed for the β-position (conducted with **3c**). Moreover, we found first-order kinetics with respect to the alkene, and zeroth-order kinetics with respect to the thiol co-catalyst, further supporting that the initial HAT to the alkene is the rate-determining step (conducted with **1a** and **1j**; Supplementary Information). Stern–Volmer quenching studies confirmed the initial oxidation of the phosphine by the excited Ir photocatalyst, and quantum yield measurements revealed that a radical chain reaction is not in operation. Overall, all these experiments and the calculations support the suggested mechanism depicted in Fig. 2b.

In summary, in situ-generated PR$_3$−OH radicals have been shown to react as HAT reagents with various alkenes and naphthalenes to give the corresponding H atom addition adduct radicals at room temperature. The key PR$_3$−OH radicals, which can be considered as formal 'free' H atoms, are readily generated through photocatalytic phosphine-mediated water activation. Thiols are used as catalytic co-reductants for the radical hydrogenation of π systems, transforming the initially generated PR$_3$−OH$_2$ radical cation intermediate, which is an efficient proton donor, into a radical hydrogen atom donor. This co-catalysis approach ensures that both H atoms of water can be used as H atom donors in the reduction of π systems. Considering hydrogenation of alkenes, the introduced metal-free PR$_3$−OH intermediate has comparable reactivity to the established transition metal-based systems, which engage in MHAT reactions. Intra- or intermolecular hydrogen atom transfer to reactive carbon- or heteroatom-centred

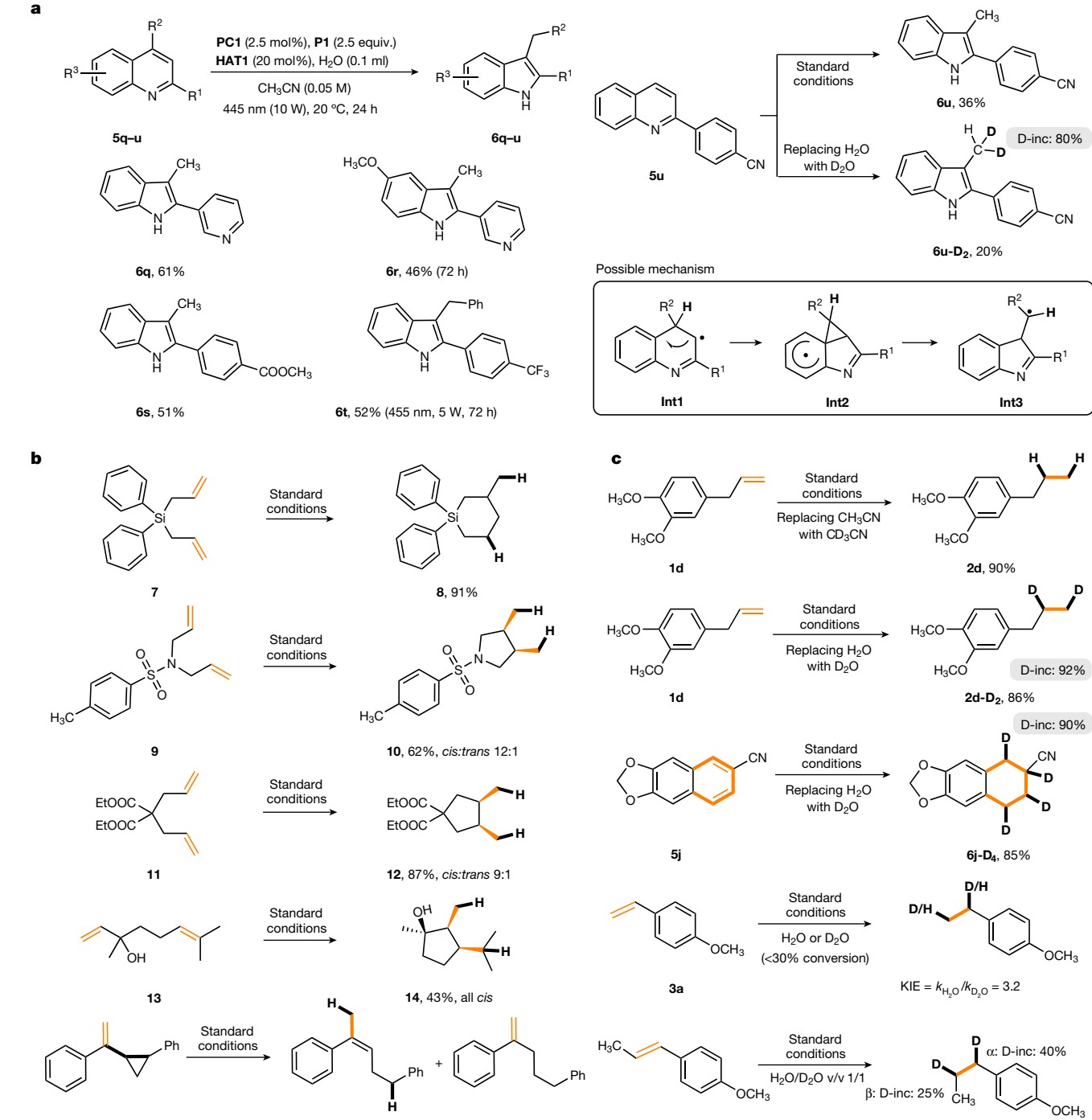

**Fig. 5 | Applications and mechanistic studies. a**, Skeletal editing from quinolines to indoles by photocatalytic phosphine-mediated H atom transfer. **b**, Radical cyclization and ring-opening experiments. **c**, Transfer hydrogenation with deuterated water to prepare deuterated reduced products. Kinetic isotope effect (KIE) measurement.

radicals is a highly useful tool in modern synthesis for radical translocation. However, HAT to closed-shell systems has not been well developed and we are confident that the strategy presented here will open the door to unexplored 'H atom radical chemistry'.

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

## Data availability

The data supporting the findings of this study are available within the paper and its Supplementary Information (experimental procedures, DFT calculations and characterization data).

**Acknowledgements** We thank the Alexander von Humboldt Foundation (J.Z.) for supporting this work. We thank T. Brake for measurement of hydrogen gas evolution. We thank T. Drennhaus for helping to analyse NMR spectra.

**Author contributions** J.Z. and A.S. conceptualized the work. J.Z. and A.S. conceived and designed the experiments. J.Z. performed the experiments and analysed the data. J.Z. and A.S. wrote the manuscript. C.M.-L. performed the density functional theory calculations on the reaction mechanism.

**Funding** Open access funding provided by Westfälische Wilhelms-Universität Münster.

**Competing interests** The authors declare no competing interests.

**Additional information**
**Correspondence and requests for materials** should be addressed to Armido Studer.
