## [Peer Review File · Nature]

Manuscript Title: Photocatalytic phosphine-mediated water activation for radical hydrogenation

Reviewer Comments & Author Rebuttals

Reviewer Reports on the Initial Version:

Referees' comments:

Referee #1 (Remarks to the Author):

This paper has a special quality that I often associate with synthetic chemistry papers published in *Nature*, leaving me with a feeling of admiration for a very cleverly designed and well executed concept, tinged with minor regret that I did not think of this idea myself! The power of phosphorus radical cations to access other reactive radicals has been known for decades, and synthetic applications have exploded in recent years (e.g. reviewed in reference 43), most noticeably following Doyle et al's study based on alcohol and carboxylic acid derived phosphorus radical cations (reference 44). To the best of my knowledge, I know of no other study in which a water-phosphorus radical cation adduct has been used to promote useful synthetic chemistry on another system [although Pandey et al did show that the key reaction design idea outlined in Figure 1d is viable, albeit in a much simpler setting, see: Pandey, G.; Pooranchand, D.; Bhalerao, U. T. Photoinduced single electron transfer activation of organophosphines: Nucleophilic trapping of phosphine radical cation. *Tetrahedron* 1991, 47, 1745–1752]. I think that this Pandey study should be cited, and briefly mentioned. But notably, this study does not go on to use the reactive intermediate in a synthetically useful way. Harnessing it to do hydrogenation transforms this knowledge from a mechanistic curiosity into a very exciting new synthetic methodology with much potential. I can certainly imagine the ready access to H radical equivalents this work enables will propagate much follow up research from groups around the world, using it in the manner designed, and also in various other radical cascade processes. Reinvention of hydrogenation will surely also help ensure that the manuscript generate very high levels of interest in the synthetic community. The conclusions and proposed mechanism are all in line with data obtained to the best of my knowledge, and the mechanism proposed looks to be reasonable (with an alternative suggestion to consider noted below).

In summary, I like this work a lot, and I would be happy see it published in *Nature*. Revisions are suggested below to be considered:

Suggested revisions that may require additional experiments:

1) While I agree that the proposed mechanism is reasonable based on the data, did the authors consider the possibility of chain reactions that do not require the Ir photocatalyst throughout? For example, it is known that thiyl radicals can react directly with phosphines to form R3P-SR radicals. I do not know whether such an intermediate might lead to further productive reactivity (i.e. reaction with water to regenerate the thiol and the key R3P-OH radical?), but this could be commented on, if the authors think this idea has merit and/or they have any insight into this point. Or were any other alternative mechanistic pathways considered and ruled out? Quantum yield experiments may shed additional light on the question of whether the Ir catalyst is involved throughout the reaction, or just as an initiator. At the end of the mechanism section, the authors wrote 'Overall, all these experiments and the calculations support the suggested mechanism depicted in Figure 2b.' I found this statement to be a bit too brief, and think that the authors could have done more to help explain why this is the case, as well as explaining why other pathways were ruled out - or if they can't be ruled out, presented as possible alternatives.

2) Scale – apologies if I missed anything, but all of the examples I checked looked to have been performed on the 10's of mg scale. This is fine for substrate scoping studies. However, can the reaction be scaled up further? For the reaction to be considered to be a genuinely useful alternative to more traditional hydrogenation methods, I feel that it should be demonstrated on gram scale for a few substrates.

3) In Scheme 4a, the authors show some examples to document an advantage of radical hydrogenation over other methods using H₂ gas. I agree that the experiments presented support this. However, an experiment not included, that I think would further enhance this point, would be to examine a diene starting material with markedly different alkenes to explore the chemoselectivity of the reaction. More specifically, it would be interesting to know whether it is possible to hydrogenate an activated alkene in the presence of an unactivated alkene, by taking advantage of radical stability in the respective intermediates.

Minor points:

Figure 2C – the spin density picture is overlapping the Chemdraw text a little.

In the naphthalene products section (Fig 3c) the reaction schemes blend in a bit too well with the products – in my opinion, drawing a light box around the reaction schemes to group them more clearly would help

Referee #3 (Remarks to the Author):

"Radical hydrogenation through hydrogen atom transfer to closed-shell π systems enabled by photocatalytic phosphine-mediated water activation"

Overview

The authors report an approach for H₂O activation that involves forming a PAr₃-OH radical intermediate that they assert behaves like a free hydrogen atom, with phosphine oxide formation being the thermodynamic driving force for this process. To illustrate this concept, they show that photoredox-catalyzed alkene hydrogenation reactions and naphthalene dearomatization reactions are viable. Both hydrogen atoms derive from water in the hydrogenation reaction, and the authors point out its abundance in the manuscript's introduction. They also detail existing processes for water activation and explain how this approach is distinct, particularly in using main group elements. Calculations in this manuscript support their premise of reducing the O–H bond strength, and, remarkably, it can decrease to as low as 5 kcal/mol depending on the phosphine employed in the reaction. Both reactions described have a reasonable scope, perhaps not what one could expect, considering that the O–H bond strength is so low and the final part of the paper contains mechanistic experiments that validate the presence of carbon-centered radicals and support the assertion that H-atoms in the final products derive from the water via D-incorporation studies.

Critique

There are several pertinent considerations when evaluating this manuscript, but they can be summarized in the following two points.

1) Use of water as a reagent vs. forming triarylphosphine byproducts.

Activating water and incorporating its hydrogen atoms into organic molecules is undoubtedly

valuable because of its abundance and potential for sustainable synthesis. To be fair to the authors, they did not overhype this aspect of the paper, but it is important to consider here, particularly because hydrogenation involving H₂, and other H₂ equivalents, are well developed. Forming triarylphosphine byproducts here is clearly a disadvantage, and I am ok with this deficiency, provided the output products are valuable and would be challenging to obtain using existing chemical methods. In my opinion, it is hard to make a case for the approach in this paper, as the hydrogenation reactions are accomplishable via other methods, and this phosphine mediate process incurs too much technical debt based on the phosphine oxide byproduct. The naphthalene reduction reaction is more novel but not a particularly groundbreaking process. I will address the novelty of the process in the next paragraph, and place more importance on that aspect, but it is a shame that the output of this discovery was not more compelling.

2) Novelty of the phosphine-mediated process and its impact on H₂O activation and HAT-mediated reaction.

This aspect is the most important part of the manuscript from my point of view. The authors do a good job of outlining the current approaches to H₂O activation that enable the transfer of its H-atoms to organic compounds, and there are certainly some viable methods out there to accomplish that feat. The manuscript narrows into water activation by main group elements as the most novel feature, and there is no doubt that this approach is unique. Although not using a main group element, the most comparable systems mechanistically are the Sm reactions, and even at this initial stage, it appears this phosphine-mediated approach is more convenient and has the potential for broader application. Forming a phosphine radical cation and then intercepting with water to form the PAr₃-OH radical is a really clever way to weaken OH bonds and was by far the most interesting part of the study. As mentioned above, it is staggering that the bond strength decreases to 5 kcal/mol. If they had matched up this phenomenon with a reaction that is unambiguously difficult for other metal hydride reduction reactions, I would have been much more excited about the promise of this finding. The hydrogenation reaction is limited in scope and starts to struggle with disubstituted systems. That is what I feel this paper is missing, a clear demonstration that the O-H activation process can be leveraged to achieve a HAT reaction that advances this area. In fact, the last sentence of the paper contains this sentiment: "However, HAT to closed-shell systems has not been well developed and we are confident that the strategy presented herein will open a door to unexplored "H atom radical chemistry"". The naphthalene reductions and radical cyclization reactions in Figure 4 are not compelling enough to illustrate that point.

Weighing up these two discussion points, I feel the paper falls just short of warranting publication in Nature. I think this O-H activation mode is an interesting discovery, and if the authors had shown a more valuable process that capitalizes on the weak O-H bond, my decision would have been reversed. I do hope that they can find such an application, as it would be a shame if the work does not go beyond the phenomenological.

Other comments

The supporting information is in excellent shape

Referee #4 (Remarks to the Author):

Professor Studer and coworkers present an interesting and potentially very useful approach for hydrogen atom transfer from water coordinated phosphine enabled by initial photocatalytic electron transfer. The utility of the approach is demonstrated through the reduction of a series of activated and unactivated alkenes and arenes. Replacement of water by deuterium oxide leads to selective reductive deuteration of alkenes. Hydrogen atom transfer by the the phosphine-water complex is demonstrated through radical cyclization ring-opening experiments and computational

studies that are fully consistent with the mechanistic hypothesis. There are a few smaller issues that can easily be addressed. For instance, some of the references do not align with the presentation and several relevant references related to the work are missing. Also, some key differences in substrate reactivity are not explored. I think this approach is potentially useful, but there are a number of issues that need to be addressed before the work is suitable for Nature.

Although HAT from from a phosphine-water complex initiated by photocatalysis is novel, the basic concept of photocatalytic bond-weakening from a main group-water complex is not. A classic recent example of the use of boronic acid-water complex to generate carbon radicals using photocatalysis was published by Bloom and coworkers in ACS Catalysis (ACS Catal. 2020, 10, 12727). I recognize that this is not HAT, but the general concept is similar producing a carbon free radical instead of a hydrogen atom to initiate free-radical reductions and bond-forming reactions.

In reading through the references, I was somewhat puzzled by the inclusion of references 18-20 as it relates to the homolytic cleavage of O-H bonds in a Sm-water complex. Also, some key references are missing. I read the references carefully and have the following comments: 1) reference 18 is a general review of reactions of samarium(II) Iodide. There is no mention of bond-weakening, HAT, or homolytic cleavage as described in the present manuscript. Reference 19 discusses the reduction and reductive coupling of barbituates by samarium diiodide-water complexes, but clearly presents them as electron transfer-proton transfer processes (ET-PT), not hydrogen atom transfer (HAT) or bond-weakening processes as described in the present manuscript. Reference 20 also describes reductions of arenes as rate-limiting electron transfers describing proton transfer as occurring after a rate-limiting steps and misinterprets isotope effects to come to this conclusion. It clearly does not present reaction as homolytic O-H bond cleavage as suggested on line 36 of the present manuscript. The first description of HAT or PCET from a samarium diiodide-water complex was presented by Flowers in 2015 (J. Am. Chem. Soc. 2015, 137, 11526-11531) and two followup articles show that reduction of carbonyls by samarium diiodide-water occur through formal HAT (J. Am. Chem. Soc. 2016, 138, 8738-8741; J. Am. Chem. Soc. 2018, 140, 15342-15352). A review on formal HAT through PCET in reductions of samarium diiodide-water complexes was published in 2019 (Dalton Trans. 2019, 48, 16142-16147). In my view, these are more relevant references since they are directly related to the concept described in the present manuscript.

In addition to citation 25 describing trialkyl borane activation of water, the authors should add the following reference which clarifies the mechanism of the process described in the Wiberg and Wood system: J. Am. Chem. Soc. 2018, 140, 155–158.

One of the shortcomings of the present work is that the system suffers from the main drawback of low-valent metal-water systems in that a stoichiometric or superstoichiometric amount of reagent to coordinate water is required for HAT. The driving force in both cases is the formation of a strong metal-oxygen bond after oxidation which in this case is the formation of the P-O bond. While I recognize that phosphine is oxidized by a photocatalysis, it's use in stoichiometric amounts doesn't differentiate it from low-valent water activation systems such as Ti, Sm, etc. In addition, given the high MW (~350) of the sacrificial phosphines used, it is not an atom-efficient reaction. The distinction between the present system and those that employ low-valent metals is that the O-H bond-weakening in the latter cases occurs through an inner-sphere process whereas the present system is proposed to occur through a sequential oxidation-coordination-HAT. Regardless of the approach employed, none of these systems deliver a hydrogen atom efficiently (or sustainably) since a sacrificial large MW complex is used to transfer a hydrogen atom.

In examining the range of reductions and reactions in Figure 3, none of the reactions are really surprising or distinctive. While I agree there are a broad range of substrates examined, I don't see this as being distinctive since there are no competing functional groups that can be reduced through HAT. Selective reduction of 2s, 2t, and 2u is unsurprising since there are no other competing functional groups that one would expect to be reduced through this method. I fully

agree with the authors statement in line 131 about the challenge of reducing arenes with by metal hydrides, but reduction by low-valent metal water complexes through formal HAT (but likely concerted PCET) is known. Reference 37 in the present manuscript provides an example, and there is a recent chemrxiv that has demonstrated naphthalene reduction that has been cited in the peer-reviewed literature (ChemRxiv preprint:2022-d9s6p, 2022). Beyond this, there are some selectivity differences in naphthalene reductions that are not addressed. For instance, in some cases, the substituted ring of the Naphthalene is reduced. In other cases, the unsubstituted portion of the ring is reduced. What is the basis of the selectivity? Why do some substrates provide excellent yields whereas others are modest?

The use of radical probes in Figure 4 is fine, but these are standard tools used to identify the presence of radicals and provide products that are fully consistent with the proposed mechanism and the mechanistic studies described at the end of the manuscript. In my view, the most impressive reaction of practical value is the conversion of 13 to 14 suggesting that there is selective reduction of the more highly substituted alkene leading to cyclization. From a synthetic standpoint, this is quite useful. In addition, the use of D₂O is useful as a deuterium source in the reductions.

One set of experiments the authors should consider is competition studies between various substituted alkenes or arenes to determine relative rates of substrate reduction. Such studies would demonstrate selectivity critical for more complex, multifunctional substrates.

If these issues are addressed and additional experiments considered as described above, this manuscript has the potential to be suitable for Nature.

Author Rebuttals to Initial Comments:

Dear Bryden,

Thank you very much for sending us your decision letter and the reviewers' comments. Based on their requests, we have revised the manuscript and the Supplementary Materials carefully with all changes made in the revised manuscript highlighted in yellow. A point-by-point response to the reviewers' comments is given below. We could address all requests raised by the three reviewers and hope that the revised version will become suitable for publication in Nature. We wish to dedicate this paper on the occasion of the 70th birthday of Prof. Dennis P. Curran. I am looking forward to hearing from you soon.

With best wishes,

Armido

Referee #1 (Remarks to the Author):

This paper has a special quality that I often associate with synthetic chemistry papers published in Nature, leaving me with a feeling of admiration for a very cleverly designed and well executed concept, tinged with minor regret that I did not think of this idea myself! The power of phosphorus radical cations to access other reactive radicals has been known for decades, and synthetic applications have exploded in recent years (e.g. reviewed in reference 43), most noticeably following Doyle et al's study based on alcohol and carboxylic acid derived phosphorus radical cations (reference 44). To the best of my knowledge, I know of no other study in which a water-phosphorus radical cation adduct has been used to promote useful synthetic chemistry on another system [although Pandey et al did show that the key reaction design idea outlined in Figure 1d is viable, albeit in a much simpler setting, see: Pandey, G.; Pooranchand, D.; Bhalerao, U. T. Photoinduced single electron transfer activation of organophosphines: Nucleophilic trapping of phosphine radical cation. *Tetrahedron* 1991, 47, 1745–1752]. I think that this Pandey study should be cited, and briefly mentioned. But notably, this study does not go on to use the reactive intermediate in a synthetically useful way. Harnessing it to do hydrogenation transforms this knowledge from a mechanistic curiosity into a very exciting new synthetic methodology with much potential. I can certainly imagine the ready access to H radical equivalents this work enables will propagate much follow up research from groups around the world, using it in the manner designed, and also in various other radical cascade processes. Reinvention of hydrogenation will surely also help ensure that the manuscript generate very high levels of interest in the synthetic community. The conclusions and proposed mechanism are all in line with data obtained to the best of my knowledge, and the mechanism proposed looks to be reasonable (with an alternative suggestion to consider noted below).

In summary, I like this work a lot, and I would be happy to see it published in Nature. Revisions are suggested below to be considered:

Reply: Thank you very much for your nice comments and suggestions. We have cited the Pandey work in the revised manuscript as new ref. 51 and made the following comment in the main text "The PR_3-H_2O radical cation can be deprotonated generating a PR_3-OH radical intermediate in a heterolytic O–H bond cleavage, as previously suggested by Pandey and co-workers.⁵¹".

Suggested revisions that may require additional experiments:

1) While I agree that the proposed mechanism is reasonable based on the data, did the authors consider the possibility of chain reactions that do not require the Ir photocatalyst throughout? For example, it is known that thiyl radicals can react directly with phosphines to form R_3P-SR radicals. I do not know whether such

an intermediate might lead to further productive reactivity (i.e reaction with water to regenerate the thiol and the key R₃P-OH radical?), but this could be commented on, if the authors think this idea has merit and/or they have any insight into this point. Or were any other alternative mechanistic pathways considered and ruled out? Quantum yield experiments may shed additional light on the question of whether the Ir catalyst is involved throughout the reaction, or just as an initiator. At the end of the mechanism section, the authors wrote ‘Overall, all these experiments and the calculations support the suggested mechanism depicted in Figure 2b.’ I found this statement to be a bit too brief, and think that the authors could have done more to help explain why this is the case, as well as explaining why other pathways were ruled out - or if they can’t be ruled out, presented as possible alternatives.

Reply: We fully agree with the referee that thiyl radicals can react with phosphines. The quantum yield, as requested, was measured and calculated to be very low, showing that a chain reaction is not operative (see SI). Hence, the Ir-catalyst is a true catalyst and not an initiator in this transformation. Furthermore, we ran Stern-Volmer quenching studies that support the initial SET-oxidation of the phosphine by the Ir-catalyst. Overall, the DFT-calculations, radical probe experiments and also deuteration experiments support the suggested mechanism. In the final sentence we state that the mechanism provided is a suggested mechanism. Furthermore, we found that a sterically highly hindered aryl thiol has to be used as catalyst. We feel that for such a sterically highly hindered thiyl radical, addition to the phosphine should be very slow.

2) Scale – apologies if I missed anything, but all of the examples I checked looked to have been performed on the 10’s of mg scale. This is fine for substrate scoping studies. However, can the reaction be scaled up further? For the reaction to be considered to be a genuinely useful alternative to more traditional hydrogenation methods, I feel that it should be demonstrated on gram scale for a few substrates.

Reply: We fully agree with the referee and performed three different gram-scale experiments with substrates **1b**, **3d** and **5c**. All larger scale experiments gave comparable yields to those obtained for the hydrogenations at smaller scale. These results were added to the revised manuscript and SI (see Figure R1).

Figure R1. Gram-scale experiments.

3) In Scheme 4a, the authors show some examples to document an advantage of radical hydrogenation over other methods using H₂ gas. I agree that the experiments presented support this. However, an experiment not included, that I think would further enhance this point, would be to examine a diene starting material with markedly different alkenes to explore the chemoselectivity of the reaction. More specifically, it would be interesting to know whether it is possible to hydrogenate an activated alkene in the presence of an unactivated alkene, by taking advantage of radical stability in the respective intermediates.

Reply: Thank you for your valuable suggestions. We ran a series of experiments along those lines and added the following text to the revised manuscript: “We next explored the relative reactivity of two different π -systems either by installing them into the same substrate (Figure 4b) or by running competition experiments of two π -systems at low conversion (see the Supplementary Materials). The alkene moiety in 2-vinylnaphthalene (**3n**) is more reactive than the naphthalene core and **4n** was obtained with complete chemoselectivity using PPh₃ (**P3**). The same outcome was noted for the more reactive phosphine **P1** (see the Supplementary Materials). Comparing an activated with an unactivated terminal alkene (**1y**), we found excellent chemoselectivity for the hydrogenation of the styrenic double bond with the milder **P3** as mediator and **2y** was obtained in 82% yield. However, both double bonds in **1y** were reduced with the more reactive phosphine **P1**. Thus, the chemoselectivity of the hydrogenation of **1y** can be controlled by H-atom reactivity tuning through variation of the PAr₃ component. This is reminiscent to the reactivity tuning in organometallic hydrides through ligand variation. Considering styrene pairs that slightly differ in their electronics by variation of their para-substituents, low selectivity was noted using the milder PPh₃-derived H-donor (Me vs. CN = 1:1.4; Me vs. MeO = 1:2). Steric effects play a role as β -methyl (trans) and also α -methyl-substituted styrenes are less reactive than their corresponding unsubstituted styrenes (around 2:1).” Additional competition experiments are provided in Figure R2 and are also provided in the SI. As indicated by the reviewer, radical stability of the H-atom-adduct plays a role. Considering HAT to naphthalenes, we ran a series of competition experiments (Figure R2). Only in the case of 2-cyanonaphthalene versus naphthalene, complete chemoselectivity for the hydrogenation of cyanonaphthalene was obtained. 1-Naphthol was around 1.8 times more reactive than naphthalene. 2-Cyanonaphthalene was 3.6 times more reactive than 1-methoxycarbonylnaphthalene.

Figure R2. Competition experiments and chemoselective hydrogenation.

“The regioselectivity issue was further investigated by DFT calculations (see the Supplementary Materials). For 1-methylnaphthalene (**5f**) the activation barrier for the intermolecular HAT from the **P1-OH** radical to the 2-, 3-, 4-, 5-, 6-, 7-, and 8-position was calculated. All HATs are exothermic (-17.9 to -23.5 kcal mol⁻¹) and occur with low barriers (6.3 to 9.7 kcal mol⁻¹). The lowest barrier was calculated for the HAT to the C4-position (6.3 kcal mol⁻¹), but H-transfer to C8 and C5 leading to the other regioisomer show similar barriers (6.5 and 6.9 kcal mol⁻¹, respectively). This explains why no selectivity was obtained for the hydrogenation of **5f**. The barrier for the HAT correlates well with the thermodynamic stability of the H-adduct (exothermicity, C4-position: -23.5 kcal mol⁻¹, C8-position: -23.5 kcal mol⁻¹, C5-position: -23.0 kcal mol⁻¹). With this knowledge in hand, we also calculated the exothermicity for the HAT to 1-(methoxycarbonyl)naphthalene (**5b**) and 2-cyanonaphthalene (**5c**), where perfect regiocontrol was achieved in the hydrogenation. Again, 7 regioisomeric H-adducts were considered for both cases. For the ester **5b**, HAT to the C4-position is thermodynamically most favored (-26.2 kcal mol⁻¹) leading to the experimentally observed regioisomer **6b**. The thermodynamically most stable H-adduct in the reaction with **5c** is resulting from C1-addition (-27.0 kcal mol⁻¹) which eventually leads to the observed isomer **6c**, revealing that the regioselectivity in these hydrogenations is mainly controlled by the stability of the dearomatized H-adduct radical.”

Figure R3a-c provides an overview of these calculations. This discussion was included into the revised version of the manuscript.

Figure R3a. HAT to 1-methylnaphthalene ($[\Delta G_{298}^{\ddagger}/\text{kcal mol}^{-1}]$, PWPB95-D3/def2-TZVP//PBE0-D3/def2-TZVP + COSMO-RS(CH_3CN)).

Figure R3b. HAT to 1-(methoxycarbonyl)naphthalene.

Figure R3c. HAT to 2-cyanonaphthalene.

Minor points:

Figure 2C – the spin density picture is overlapping the Chemdraw text a little.

In the naphthalene products section (Fig 3c) the reaction schemes blend in a bit too well with the products – in my opinion, drawing a light box around the reaction schemes to group them more clearly would help

Reply: Thanks, we have updated the Figure 2c and now use light boxes around the reaction schemes in the revised Figure 3.

Referee #3 (Remarks to the Author):

"Radical hydrogenation through hydrogen atom transfer to closed-shell π systems enabled by photocatalytic phosphine-mediated water activation"

Overview

The authors report an approach for H₂O activation that involves forming a PAr₃-OH radical intermediate that they assert behaves like a free hydrogen atom, with phosphine oxide formation being the thermodynamic driving force for this process. To illustrate this concept, they show that photoredox-catalyzed alkene hydrogenation reactions and naphthalene dearomatization reactions are viable. Both hydrogen atoms derive from water in the hydrogenation reaction, and the authors point out its abundance in the manuscript's introduction. They also detail existing processes for water activation and explain how this approach is distinct, particularly in using main group elements. Calculations in this manuscript support their premise of reducing the O–H bond strength, and, remarkably, it can decrease to as low as 5 kcal/mol depending on the phosphine employed in the reaction. Both reactions described have a reasonable scope, perhaps not what one could expect, considering that the O–H bond strength is so low and the final part of

the paper contains mechanistic experiments that validate the presence of carbon-centered radicals and support the assertion that H-atoms in the final products derive from the water via D-incorporation studies.

Reply: Thank you for your nice comments!

Critique

There are several pertinent considerations when evaluating this manuscript, but they can be summarized in the following two points.

1) Use of water as a reagent vs. forming triarylphosphine byproducts.

Activating water and incorporating its hydrogen atoms into organic molecules is undoubtedly valuable because of its abundance and potential for sustainable synthesis. To be fair to the authors, they did not overhype this aspect of the paper, but it is important to consider here, particularly because hydrogenation involving H₂, and other H₂ equivalents, are well developed. Forming triarylphosphine byproducts here is clearly a disadvantage, and I am ok with this deficiency, provided the output products are valuable and would be challenging to obtain using existing chemical methods. In my opinion, it is hard to make a case for the approach in this paper, as the hydrogenation reactions are accomplishable via other methods, and this phosphine mediate process incurs too much technical debt based on the phosphine oxide byproduct. The naphthalene reduction reaction is more novel but not a particularly groundbreaking process. I will address the novelty of the process in the next paragraph, and place more importance on that aspect, but it is a shame that the output of this discovery was not more compelling.

Reply: Thank you for your suggestions. We agree that the phosphine oxide byproduct formation might hamper larger scale hydrogenation using this methodology. However, the phosphine oxide could be almost quantitatively recovered (Figure R4). It is known that phosphine oxides can be reduced to give the corresponding phosphines (Chem. Soc. Rev. 2015, 44, 2508) with entries found also in the patent literature (BASF). Therefore, this shortcoming may be addressed through phosphine recycling after the hydrogenation. Further, it should be possible to develop a method for the in-situ reduction, but this would go beyond the scope of this initial paper.

Figure R4. Gram-scale experiments with recycling of the phosphine oxide byproduct.

2) Novelty of the phosphine-mediated process and its impact on H_2O activation and HAT-mediated reaction.

This aspect is the most important part of the manuscript from my point of view. The authors do a good job of outlining the current approaches to H_2O activation that enable the transfer of its H-atoms to organic compounds, and there are certainly some viable methods out there to accomplish that feat. The manuscript narrows into water activation by main group elements as the most novel feature, and there is no doubt that this approach is unique. Although not using a main group element, the most comparable systems mechanistically are the Sm reactions, and even at this initial stage, it appears this phosphine-mediated approach is more convenient and has the potential for broader application. Forming a phosphine radical cation and then intercepting with water to form the $\text{PAR}_3\text{-OH}$ radical is a really clever way to weaken OH bonds and was by far the most interesting part of the study. As mentioned above, it is staggering that the bond strength decreases to 5 kcal/mol. If they had matched up this phenomenon with a reaction that is unambiguously difficult for other metal hydride reduction reactions, I would have been much more excited about the promise of this finding. The hydrogenation reaction is limited in scope and starts to struggle with disubstituted systems. That is what I feel this paper is missing, a clear demonstration that the O–H activation process can be leveraged to achieve a HAT reaction that advances this area. In fact, the last sentence of the paper contains this sentiment: "However, HAT to closed-shell systems has not been well developed and we are confident that the strategy presented herein will open a door to unexplored "H atom radical chemistry"". The naphthalene reductions and radical cyclization reactions in Figure 4 are not compelling enough to illustrate that point.

Reply: Thank you for your suggestions and for your advice to further improve the quality of the paper. To document an additional advantage of this hydrogenation method over state-of-the-art hydrogenation chemistry (transition metal-based systems or radical methodology), we realized a hydrogenation that proceeds with concomitant skeletal editing, harvesting the reactivity of radical intermediates in our processes. Thus, quinolines could be reductively directly converted to indoles by our photocatalytic phosphine-mediated H-atom transfer. Several indole derivatives could be constructed through a ring contraction of the corresponding quinoline derivatives (see Figure R5). The rearrangement of 2-substituted quinolines (**5q-s** and **5u**) gave the corresponding 2,3-disubstituted indoles (**6q-s** and **6u**) and a 2,4-disubstituted quinoline (**5t**) also engaged in this hydrogenative skeletal editing. The suggested mechanism of the cascade was supported by a deuteration experiment upon replacing H_2O by D_2O . These in our eyes exciting new results were added to the manuscript. To the best of our knowledge such a hydrogenative skeletal editing cannot be achieved with any hydrogenation method known in the literature. We hope that these experiments are compelling enough to illustrate the potential of our novel approach to develop unexplored H-atom radical chemistry.

Figure R5. Single-atom skeletal editing from quinolines to indoles by photocatalytic phosphine-mediated H-atom transfer.

Weighing up these two discussion points, I feel the paper falls just short of warranting publication in Nature. I think this O–H activation mode is an interesting discovery, and if the authors had shown a more valuable process that capitalizes on the weak O–H bond, my decision would have been reversed. I do hope that they can find such an application, as it would be a shame if the work does not go beyond the phenomenological.

Reply: Thank you for helping us to further improve the quality of this manuscript.

Other comments

The supporting information is in excellent shape

Reply: Thank you for your positive comment on the SI.

Referee #4 (Remarks to the Author):

Professor Studer and coworkers present an interesting and potentially very useful approach for hydrogen atom transfer from water coordinated phosphine enabled by initial photocatalytic electron transfer. The utility of the approach is demonstrated through the reduction of a series of activated and unactivated alkenes and arenes. Replacement of water by deuterium oxide leads to selective reductive deuteration of alkenes. Hydrogen atom transfer by the phosphine-water complex is demonstrated through radical cyclization ring-opening experiments and computational studies that are fully consistent with the mechanistic hypothesis. There are a few smaller issues that can easily be addressed. For instance, some of the references do not align with the presentation and several relevant references related to the work are missing. Also, some key differences in substrate reactivity are not explored. I think this approach is potentially useful, but there are a number of issues that need to be addressed before the work is suitable for Nature.

Reply: Thank you for your positive comments.

Although HAT from a phosphine-water complex initiated by photocatalysis is novel, the basic concept of photocatalytic bond-weakening from a main group-water complex is not. A classic recent example of the use of boronic acid-water complex to generate carbon radicals using photocatalysis was published by Bloom and coworkers in ACS Catalysis (ACS Catal. 2020, 10, 12727). I recognize that this is not HAT, but the general concept is similar producing a carbon free radical instead of a hydrogen atom to initiate free-radical reductions and bond-forming reactions.

Reply: Thank you for your suggestion. We have added that important reference in the updated manuscript as ref. 17.

In reading through the references, I was somewhat puzzled by the inclusion of references 18-20 as it relates to the homolytic cleavage of O-H bonds in a Sm-water complex. Also, some key references are missing. I read the references carefully and have the following comments: 1) reference 18 is a general review of reactions of samarium(II) Iodide. There is no mention of bond-weakening, HAT, or homolytic cleavage as described in the present manuscript. Reference 19 discusses the reduction and reductive coupling of barbituates by samarium diiodide-water complexes, but clearly presents them as electron transfer-proton transfer processes (ET-PT), not hydrogen atom transfer (HAT) or bond-weakening processes as described in the present manuscript. Reference 20 also describes reductions of arenes as rate-limiting electron transfers describing proton transfer as occurring after a rate-limiting steps and misinterprets isotope effects to come to this conclusion. It clearly does not present reaction as homolytic O-H bond cleavage as suggested on line 36 of the present manuscript. The first description of HAT or PCET from a samarium diiodide-water complex was presented by Flowers in 2015 (J. Am. Chem. Soc. 2015, 137, 11526-11531) and two followup articles show that reduction of carbonyls by samarium diiodide-water occur through formal HAT (J. Am. Chem. Soc. 2016, 138, 8738-8741; J. Am. Chem. Soc. 2018, 140, 15342-15352). A review on formal HAT through PCET in reductions of samarium diiodide-water complexes was published in 2019 (Dalton Trans. 2019, 48, 16142-16147). In my view, these are more relevant references since they are directly related to the concept described in the present manuscript.

Reply: Thank you for your suggestions. Accordingly, we have deleted refs. 18-20 and added the important references mentioned as refs. 20-23 in the revised manuscript.

In addition to citation 25 describing trialkyl borane activation of water, the authors should add the following reference which clarifies the mechanism of the process described in the Wiberg and Wood system: J. Am. Chem. Soc. 2018, 140, 155–158.

Reply: Thank you for your suggestion. We have added that JACS paper which appears in the revised manuscript as ref. 29.

One of the shortcomings of the present work is that the system suffers from the main drawback of low-valent metal-water systems in that a stoichiometric or superstoichiometric amount of reagent to coordinate water is required for HAT. The driving force in both cases is the formation of a strong metal-oxygen bond after oxidation which in this case is the formation of the P-O bond. While I recognize that phosphine is oxidized by a photocatalysis, its use in stoichiometric amounts doesn't differentiate it from low-valent water activation systems such as Ti, Sm, etc. In addition, given the high MW (~350) of the sacrificial phosphines used, it is not an atom-efficient reaction. The distinction between the present system and those that employ

low-valent metals is that the O-H bond-weakening in the latter cases occurs through an inner-sphere process whereas the present system is proposed to occur through a sequential oxidation-coordination-HAT. Regardless of the approach employed, none of these systems deliver a hydrogen atom efficiently (or sustainably) since a sacrificial large MW complex is used to transfer a hydrogen atom.

Reply: Thanks for that comment. As already responded to the second reviewer, we agree that the phosphine oxide byproduct formation might hamper larger scale hydrogenation using this methodology. However, the phosphine oxide could be almost quantitatively recovered (see Figure R4). It is known that phosphine oxides can be reduced to give the corresponding phosphines (Chem. Soc. Rev. 2015, 44, 2508) with entries found also in the patent literature (BASF). Therefore, this shortcoming may be addressed through phosphine recycling after the hydrogenation. Further, it should be possible to develop a method for the in-situ reduction, but this would go beyond the scope of this initial paper. Furthermore, the phosphine entity can be used to tune the reactivity of the H-atom donor (see comment to referee 1 on chemoselective hydrogenations).

In examining the range of reductions and reactions in Figure 3, none of the reactions are really surprising or distinctive. While I agree there are a broad range of substrates examined, I don't see this as being distinctive since there are no competing functional groups that can be reduced through HAT. Selective reduction of 2s, 2t, and 2u is unsurprising since there are no other competing functional groups that one would expect to be reduced through this method. I fully agree with the authors statement in line 131 about the challenge of reducing arenes with by metal hydrides, but reduction by low-valent metal water complexes through formal HAT (but likely concerted PCET) is known. Reference 37 in the present manuscript provides an example, and there is a recent chemrxiv that has demonstrated naphthalene reduction that has been cited in the peer-reviewed literature (ChemRxiv preprint:2022-d9s6p, 2022). Beyond this, there are some selectivity differences in naphthalene reductions that are not addressed. For instance, in some cases, the substituted ring of the Naphthalene is reduced. In other cases, the unsubstituted portion of the ring is reduced. What is the basis of the selectivity? Why do some substrates provide excellent yields whereas others are modest?

Reply: Thank you for your suggestions. Based on your questions, we performed a series of experiments as shown below, these results have been added to the revised manuscript and SI:

*The novelty of this method regarding the substrate scope. As already responded to referee 3, we managed to realize a hydrogenation of quinolines with concomitant skeletal editing to give indoles. Thus, quinolines could be reductively directly converted to indoles by our photocatalytic phosphine-mediated H-atom transfer. Several indole derivatives could be constructed through a ring contraction of the corresponding quinoline derivatives (see Figure R5 above). The rearrangement of 2-substituted quinolines (**5q-s** and **5u**) gave the corresponding 2,3-disubstituted indoles (**6q-s** and **6u**) and a 2,4-disubstituted quinoline (**5t**) also engaged in this hydrogenative skeletal editing. The suggested mechanism of the cascade was supported by a deuteration experiment upon replacing H₂O by D₂O. These in our eyes exciting new results were added to the manuscript. To the best of our knowledge such a hydrogenative skeletal editing cannot be achieved with any hydrogenation method known in the literature.*

*Regarding functional group tolerance, we added five substrates bearing different functional groups, such as aldehyde (**2p**), thiophene (**2q**), ketone (**2x**), fluoride (**4b**) and chloride (**4g**) that did not interfere under the applied conditions. All these substrates worked well with complete chemoselectivity (Figure R6).*

Figure R6. New products.

Low-valent metal water complexes such as $Sm(II)-H_2O$ systems could be applied to the reduction of naphthalenes. However, dihydronaphthalene products were obtained in these cases, whereas our process delivers tetrahydronaphthalenes. As far as we see, the $Sm(II)-H_2O$ systems do not work well for the hydrogenation of unactivated alkenes. Our activation method and reaction mechanism are different from the $Sm(II)-H_2O$ systems (PCET) and the resulting $P(Ar)_3-OH$ intermediate can directly transfer a H atom to the alkene avoiding the high barrier for the electron transfer process of the related PCET mechanism in unactivated alkenes, which are difficult to be reduced through electron transfer.

Regarding the selectivity issue, we ran additional studies (see also response to referee 1). We found that it is possible to distinguish styrenic (activated) double bonds from unactivated terminal alkenes. We could show that the proper phosphine is important to achieve that chemoselectivity. With the more reactive $PAryl_3-OH$ radical intermediate (lower O-H BDE) derived from **P1**, no selectivity was obtained, whereas with **P3** complete chemoselectivity was noted (see Figure R2 above). Thus, the chemoselectivity of the hydrogenation of **Iy** can be controlled by H-atom reactivity tuning through variation of the $PAryl_3$ component. This is reminiscent to the reactivity tuning in organometallic hydrides through ligand variation. Considering styrene pairs that slightly differ in their electronics by variation of their para-substituents, low selectivity was noted using the milder PPh_3 -derived H-donor (Me vs. CN = 1:1.4; Me vs. MeO = 1:2). Steric effects play a role as β -methyl (trans) and also α -methyl-substituted styrenes are less reactive than their corresponding unsubstituted styrenes (around 2:1).” Additional competition experiments are provided in Figure R2 above and are also provided in the SI. Radical stability of the H-atom-adduct plays a role. Considering HAT to naphthalenes, we ran a series of competition experiments (Figure R2). Only in the case of 2-cyanonaphthalene versus naphthalene, complete chemoselectivity for the hydrogenation of cyanonaphthalene was obtained. 1-Naphthol was around 1.8 times more reactive than naphthalene. 2-Cyanonaphthalene was 3.6 times more reactive than 1-methoxycarbonylnaphthalene. To further address chemoselectivity for the naphthalene reductions, DFT calculations (see SI) were performed on various derivatives. We found that the experimentally obtained chemoselectivity for such hydrogenations correlates well with the radical stability of the initial H-adduct (thermochemistry, see also response to referee 1 and Figure R3).

Regarding the comment “Some substrates provide excellent yields whereas others are modest”: Most these hydrogenation reactions are clean and some of them gave a modest yield because of their lower reactivity, such as **1n**, **1t** and **3f**. As reactions are clean, unreacted starting material remained. We have further optimized and found that a slight increase of the reaction temperature to 40 °C improved the conversion for more difficult substrates. Thus, we repeated selected experiments under these slightly modified conditions and the higher yields obtained are shown in Figure R7 (in parenthesis). The yields reported in our first submission are presented for comparison. The manuscript was revised accordingly.

Figure R7. Yield comparison between the reactions conducted at 20 °C and 40 °C (in parenthesis).

The use of radical probes in Figure 4 is fine, but these are standard tools used to identify the presence of radicals and provide products that are fully consistent with the proposed mechanism and the mechanistic studies described at the end of the manuscript. In my view, the most impressive reaction of practical value is the conversion of 13 to 14 suggesting that there is selective reduction of the more highly substituted alkene leading to cyclization. From a synthetic standpoint, this is quite useful. In addition, the use of D2O is useful as a deuterium source in the reductions.

Reply: Thank you for the nice comments.

One set of experiments the authors should consider is competition studies between various substituted alkenes or arenes to determine relative rates of substrate reduction. Such studies would demonstrate selectivity critical for more complex, multifunctional substrates.

As already mentioned above, we ran additional studies (see also response to referee 1). We found that it is possible to distinguish styrenic (activated) double bonds from unactivated terminal alkenes. We could show that the proper phosphine is important to achieve that chemoselectivity. With the more reactive PAr₃ radical intermediate (lower O-H BDE) derived from P1, no selectivity was obtained, whereas with P3 complete chemoselectivity was noted (see Figure R2 above). Thus, the chemoselectivity of the hydrogenation of 1y can be controlled by H-atom reactivity tuning through variation of the PAr₃ component. This is reminiscent to the reactivity tuning in organometallic hydrides through ligand variation. Considering styrene pairs that slightly differ in their electronics by variation of their para-substituents, low selectivity was noted using the milder PPh₃-derived H-donor (Me vs. CN = 1:1.4; Me vs. MeO = 1:2). Steric effects play a role as β-methyl (trans) and also α-methyl-substituted styrenes are less reactive than their corresponding unsubstituted styrenes (around 2:1).” Additional competition experiments are provided in Figure R2 above and are also provided in the SI. Radical stability of the H-atom-adduct plays a role. Considering HAT to naphthalenes, we ran a series of competition experiments (Figure R2). Only in the case of 2-cyanonaphthalene versus naphthalene, complete chemoselectivity for the hydrogenation of cyanonaphthalene was obtained. 1-Naphthol was around 1.8 times more reactive than naphthalene. 2-Cyanonaphthalene was 3.6 times more reactive than 1-methoxycarbonylnaphthalene. To further address chemoselectivity for the naphthalene reductions, DFT calculations (see SI) were performed on various derivatives. We found that the experimentally obtained chemoselectivity for such hydrogenations correlates well with the radical stability of the initial H-adduct (thermochemistry). See comment to referee 1 and Figure R3.

If these issues are addressed and additional experiments considered as described above, this manuscript has the potential to be suitable for Nature.

***Reply:** Thank you for your comments. We hope with the additional experiments and results presented, your request could be fully addressed. Thanks for helping us to further improve the quality of this paper.*

Reviewer Reports on the First Revision:

Referees' comments:

Referee #1 (Remarks to the Author):

I previously recommended acceptance of this manuscript subject to revisions being made. I am happy to say that the authors took all of my suggested revisions seriously and addressed them in full to a high standard. The successful scale up results are especially pleasing to see. I also note further improvements to the manuscript following requested revisions from the other reviewers.

Therefore, at this point I am happy to recommend acceptance.

Dr Will Unsworth

Referee #3 (Remarks to the Author):

Revision of "Radical hydrogenation through hydrogen atom transfer to closed-shell n systems enabled by photocatalytic phosphine-mediated water activation"

In their revised manuscript, the authors have addressed the two main points that comprised my initial review. Regarding stoichiometry and waste, the authors argue that they can recover the phosphine oxide byproducts in close to quantitative yields and that there are several efficient ways now to reduce back down to the phosphine. I think this argument is valid and adds credibility to a more efficient process.

In truth, my main contention with the chemistry was that it is remarkable that an intermediate is obtainable with such a low O-H bond strength but that the reactions in the manuscript did not overly convince me that this process would lead to new type of reduction processes that: i) would not be achievable via conventional methods and ii) are reactions that the community would consider important/interesting (with this last point being subjective, of course). In their revised manuscript, they applied their phosphine-mediate reduction reactions to quinolines and saw an unusual rearrangement process resulting in substituted quinolines. This reaction is exactly what I was eluding to in my first evaluation. It is unique, novel, and something that would appear beyond what other radical reduction processes are capable of. I am not making these statements because the process falls under the umbrella of 'skeletal editing,' a current hot topic. It shows that the weak O-H bond in the PAr_3-OH radical is indeed a means to create new types of chemical reactions, and I now consider that the paper is significantly more impactful. The authors also responded well to the comments from the other reviewers, which also strengthened the manuscript.

Now, I fully support publishing this work in Nature, and I congratulate the authors on this work.

Referee #4 (Remarks to the Author):

I have reviewed the article carefully and the authors have addressed nearly all of the reviewer comments. In my view, this is a substantially improved manuscript for publication in Nature. My only reservation is the lack of a response in the manuscript to the production of phosphine oxide side product. In the response reviewers, the authors have noted that the phosphine oxide is isolated in nearly quantitative yield. The authors further note that methods are available that could be incorporated to recycle isolated phosphine oxide, but argue that this is beyond the scope of the present work. I want to be careful about being critical of this beautiful work, but I do think it would be beneficial to demonstrate the ease of recycling through the demonstration with at least one

example, or alternatively, note this challenge and cite work that supports their response. This isn't so much about demonstrating a response to reviewers where reasonable colleagues may disagree, but a demonstration that this approach is more sustainable than other processes; a point that will make this work more impactful.

Author Rebuttals to First Revision:

Referee #1 (Remarks to the Author):

I previously recommended acceptance of this manuscript subject to revisions being made. I am happy to say that the authors took all of my suggested revisions seriously and addressed them in full to a high standard. The successful scale up results are especially pleasing to see. I also note further improvements to the manuscript following requested revisions from the other reviewers. Therefore, at this point I am happy to recommend acceptance.

Dr Will Unsworth

Reply: We are grateful to the reviewer for supporting our work.

Referee #3 (Remarks to the Author):

Revision of "Radical hydrogenation through hydrogen atom transfer to closed-shell π systems enabled by photocatalytic phosphine-mediated water activation"

In their revised manuscript, the authors have addressed the two main points that comprised my initial review. Regarding stoichiometry and waste, the authors argue that they can recover the phosphine oxide byproducts in close to quantitative yields and that there are several efficient ways now to reduce back down to the phosphine. I think this argument is valid and adds credibility to a more efficient process.

In truth, my main contention with the chemistry was that it is remarkable that an intermediate is obtainable with such a low O-H bond strength but that the reactions in the manuscript did not overly convince me that this process would lead to new type of reduction processes that: i) would not be achievable via conventional methods and ii) are reactions that the community would consider important/interesting (with this last point being subjective, of course). In their revised manuscript, they applied their phosphine-mediate reduction reactions to quinolines and saw an unusual rearrangement process resulting in substituted quinolines. This reaction is exactly what I was eluding to in my first evaluation. It is unique, novel, and something that would appear beyond what other radical reduction processes are capable of. I am not making these statements because the process falls under the umbrella of 'skeletal editing,' a current hot topic. It shows that the weak O-H bond in the PAr₃-OH radical is indeed a means to create new types of chemical reactions, and I now consider that the paper is significantly more impactful. The authors also responded well to the comments from the other reviewers, which also strengthened the manuscript.

Now, I fully support publishing this work in Nature, and I congratulate the authors on this work.

Reply: We are grateful to the reviewer for the favorable comment and recommendation.

Referee #4 (Remarks to the Author):

I have reviewed the article carefully and the authors have addressed nearly all of the reviewer comments. In my view, this is a substantially improved manuscript for publication in Nature. My only reservation is the lack of a response in the manuscript to the production of phosphine oxide side product. In the response reviewers, the authors have noted that the phosphine oxide is isolated in nearly quantitative yield. The authors further note that methods are available that could

be incorporated to recycle isolated phosphine oxide, but argue that this is beyond the scope of the present work. I want to be careful about being critical of this beautiful work, but I do think it would be beneficial to demonstrate the ease of recycling through the demonstration with at least one example, or alternatively, note this challenge and cite work that supports their response. This isn't so much about demonstrating a response to reviewers where reasonable colleagues may disagree, but a demonstration that this approach is more sustainable than other processes; a point that will make this work more impactful.

Reply: We are grateful to the reviewer for the favorable comment and recommendation. We have cited a review in the revised manuscript as new ref. 46 and made the following comment in the main text "The generated phosphine oxides could be almost quantitatively recovered (see the Supplementary Materials) and are easily reduced to give the corresponding starting phosphines⁴⁶".